# Improved Single Breath-Hold SSFSE Sequence for Liver MRI Based on Compressed Sensing: Evaluation of Image Quality Compared with Conventional T2-Weighted Sequences

**DOI:** 10.3390/diagnostics12092164

**Published:** 2022-09-06

**Authors:** Hyun Kyung Lee, Ji Soo Song, Weon Jang, Dominik Nickel, Mun Young Paek

**Affiliations:** 1Department of Radiology, Jeonbuk National University Medical School and Hospital, Jeonju 54907, Korea; 2Research Institute of Clinical Medicine, Jeonbuk National University, Jeonju 54907, Korea; 3Biomedical Research Institute, Jeonbuk National University Hospital, Jeonju 54907, Korea; 4MR Application Predevelopment, Siemens Healthcare, 91052 Erlangen, Germany; 5Siemens Healthineers Ltd., Seoul 03737, Korea

**Keywords:** magnetic resonance imaging, liver, data compression, breath holding

## Abstract

The purpose of this study was to evaluate the image quality of compressed-sensing accelerated single-shot fast spin-echo (SSFSE_CS_) sequences acquired within a single breath-hold in comparison with conventional SSFSE (SSFSE_CONV_) and multishot TSE (mTSE). A total of 101 patients who underwent liver MRI at 3 T, including SSFSE_CONV_ (acquisition time (TA) = 58–62 s), mTSE (TA = 108 s), and SSFSE_CS_ (TA = 18 s), were included in this retrospective study. Two radiologists assessed the three sequences with respect to artifacts, organ sharpness, small structure visibility, overall image quality, and conspicuity of main lesions of liver and pancreas using a five-point evaluation scale system. Descriptive statistics and the Wilcoxon signed-rank test were used for statistical analysis. SSFSE_CS_ was significantly better than SSFSE_CONV_ and mTSE for artifacts, small structure visibility, overall image quality, and conspicuity of main lesions (*p* < 0.005). Regarding organ sharpness, mTSE and SSFSE_CS_ did not significantly differ (*p* = 0.554). Conspicuity of liver lesion did not significantly differ between SSFSE_CONV_ and mTSE (*p* = 0.404). SSFSE_CS_ showed superior image quality compared with SSFSE_CONV_ and mTSE despite a more than three-fold reduction in TA, suggesting a remarkable potential for saving time in liver imaging.

## 1. Introduction

For evaluation of focal liver lesions, contrast-enhanced computed tomography (CT) is the most widely used imaging modality. However, one of the main drawbacks of CT is its radiation hazard, and a constantly increasing demand for liver magnetic resonance imaging (MRI) in daily practice is inevitable. Liver MRI is usually regarded as ‘the final decision maker’ in the imaging work-up. As the main advantage over other imaging modalities, it provides multiparametric information from a variety of sequences, such as T1, T2-weighted imaging (T2WI) and diffusion-weighted imaging, and various contrast agents, including hepatocyte-specific agents. Although it is routinely used in daily clinical practice, it nevertheless remains difficult to obtain sufficient image quality as a result of the complexity in organizing data sampling and breath-hold (BH) timing within a short time window, especially for older or extremely ill patients [1,2]. Among these sequences, T2WI is one of the most difficult sequences to acquire with appropriate image quality due to motion or breathing artifacts, which reduce diagnostic confidence due to image blurring, ghosting, loss of signal intensity, and misregistration.

T2WI is routinely based on a turbo-spin-echo (TSE) sequence that uses Cartesian k-space sampling. To achieve T2WI without motion artifacts, alterations to the standard imaging technique can include free-breathing sequences, and respiratory-triggered (RT) and motion robust methods (such as BLADE, PROPELLER, or radial acquisition of k-space) [3,4,5]. Motion robust techniques such as BLADE or PROPELLER can improve motion compensation and signal-to-noise ratio (SNR); however, several BH or RT acquisitions are still required, which can result in a long acquisition time. Currently, faster acquisition times are needed to ensure more patient access to compensate for increases in patient visits.

Single-shot methods, including the single-shot fast spin-echo (SSFSE) sequence, account for the greatest motion-robust T2-weighted acquisitions [6,7]. Data from a single slice are acquired in a fraction of a second, and thus do not demonstrate motion artifacts similar to those from segmented T2-weighted sequences. Nevertheless, image quality can still suffer from T2 decay and low SNR, because k-space data are collected fast and echo trains take longer relative to segmented T2-weighted sequences. The acquisition time may be potentially faster as well, though limitations to the specific absorption rate (SAR) can require pauses that result in longer repetition times.

In the past few years, an MR technique using accelerated compressed sensing (CS) with a sparsity-based reconstruction of extremely undersampled data has been developed and has demonstrated much potential in making MR data acquisition faster [8,9]. Several recent studies have proven its clinical value in 3D MRCP sequences, enabling single BH-MRCP with comparable or even better image quality compared with conventional 3D-RT-MRCP [10,11]. Thus, we hypothesized that SSFSE combined with a CS technique (SSFSE_CS_) can produce comparable or even better image quality than that of conventional T2 sequences including multishot TSE (mTSE) and conventional SSFSE (SSFSE_CONV_) with an even shorter acquisition time.

The purpose of this study was to evaluate the feasibility of a single BH SSFSE_CS_ compared with SSFSE_CONV_ and mTSE.

## 2. Materials and Methods

### 2.1. Patients

This retrospective study was approved by the institutional review board of our hospital and informed consent was waived. From January 2021 to April 2021, 101 consecutive patients (76 men and 25 women; mean age, 63.1 years; range, 27–89 years) who underwent clinically indicated liver MRI were included. Indications for the liver MRIs were as follows: cirrhosis (*n* = 58), chronic hepatitis (*n* = 6), focal liver lesion other than HCC or metastasis (*n* = 17), liver metastasis (*n* = 12), and others (*n* = 8). Of the 101 patients, 54 patients had focal liver lesions (HCC (*n* = 26; size range, 1.3–4.1 cm), metastasis (*n* = 12; size range, 1–2.3 cm), hemangioma (*n* = 9; size range, 0.6–5.1 cm), inflammatory nodule (*n* = 4; size range, 1–1.5 cm), intrahepatic cholangiocarcinoma (*n* = 2; 1.7 and 2.1 cm), and angiomyolipoma (*n* = 1; 2.9 cm)), and 16 patients had intraductal papillary mucinous neoplasm (IPMN) of the pancreas (size range, 0.4–2.6 cm). In patients with multiple lesions, only the largest lesion was analyzed. All lesions were confirmed histopathologically or by combined interpretation of all available imaging modalities including endoscopic ultrasound, endoscopic retrograde cholangiopancreatography, and follow-up CT or MRI.

### 2.2. MR Imaging Techniques

All MR examinations were performed on a 3 T MR scanner (MAGNETOM Skyra, VE11E version, and Vida, VA20A version, Siemens Healthcare, Erlangen, Germany) using a combination of 18-channel (Skyra) and 30-channel (Vida) flexible anterior body coils and 32-channel spine coil elements for signal acquisition. All patients underwent three T2WI acquisitions before the injection of gadoxetic acid (Eovist/Primovist; Bayer Healthcare, Berlin, Germany). The T2WI comprised the following: (1) a conventional axial T2W SSFSE with fat suppression (SSFSE_CONV_), (2) a multishot T2W turbo spin-echo (TSE) with fat suppression (mTSE), and (3) an axial T2W SSFSE_CS_ with fat suppression. For SSFSE acquisition, the percentage of partial Fourier was 75% (6/8), and an additional homodyne filter was applied to synthesize the missing k-space data. In addition, the acquisition time per slice was 0.7–1.0 s in SSFSE_CONV_ and 0.5–0.6 s in SSFSE_CS_. The phase encode direction was anterior–posterior for all T2WIs. The detailed acquisition parameters are provided in Table 1.

### 2.3. Compressed Sensing Accelerated SSFSE Sequence

With the aim to further improve the conventional SSFSE sequence, the acquisition of the employed prototypical sequence was modified by using a variable-density sampling pattern with a total acceleration of 3.0 in the phase encode direction of Cartesian trajectory with a total acceleration and the increase of the acceleration from k-space center to periphery specified by the acceleration increment. Furthermore, a separate reference scan with 24 automatic calibration signal lines for the acquisition of the calibration data used in the estimation of coil sensitivity maps was selected. The latter was realized by a second echo train following the undersampled echo train for the acquisition of the imaging data. These modifications were introduced to shorten the duration of the echo train and therefore to reduce T2-decay-related blurring as well as the SAR determined by the number of refocusing pulses. Consequently, the sequence also allowed shorter repetition times, which directly translated into shorter acquisition times. To reduce crosstalk and magnetization transfer effects at shorter repetition times, a slice increment of four (instead of two, which was used for SSFSE_CONV_) for the acquisition of subsequent slices was used.

In order to improve the signal-to-noise ratio of the accelerated acquisition, a compressed sensing reconstruction was used that optimizes the cost function
(1)$$I_{CS}=\underset{I}{\min}\left(\frac{1}{2}\|AI-{D\|}_{2}^{2}+\lambda \;\|W{I\|}_{1}\right),$$
where **I** is the two-dimensional image, **D** is acquired k-space data of the imaging scan, *λ* is a regularization parameter, **W** is the Haar wavelet transformation, and **A** is the system operator consisting of multiplication with coil-sensitivity maps estimated from the reference scan, Fourier transformation, and masking. The second term enforces sparsity in the wavelet domain through the properties of the l1 norm. The reconstruction was integrated into the reconstruction pipeline on the scanner, and the CPU implementation required about 1 s/slice for the given protocol (approximately 30–40 s per patient).

### 2.4. Image Analysis

All three T2WI assessments were performed blinded and by consensus of two readers with 14 years and 7 years of abdominal MRI experience. Each reviewer was not aware of the sequence parameters or patient information. All MR images were optimally cropped in order to remove pulsation and motion artifacts that are prominent on mTSE images. Adjusting the window level and width was permitted for the qualitative assessment. The readers assessed three T2WIs with regard to artifacts of any kind (e.g., motion, pulsation, SENSE, and GRAPPA), sharpness of upper abdominal organ margin (liver, spleen, gallbladder, and pancreas), visibility of small vessel and intrahepatic duct, overall image quality, and conspicuity of the main lesion for each patient using a five-point evaluation scale system (1 = unacceptable, 2 = poor, 3 = fair, 4 = good, and 5 = excellent). Images were considered insufficient when rated 1 or 2. The study coordinator, who was not involved in image interpretation, assigned the largest lesion as the target lesion before image review for patients with multiple lesions. Each review was performed three times on days separated by a minimum of 2 weeks.

### 2.5. Statistical Analysis

Numerical values were presented as mean ± standard deviation. We also calculated descriptive statistics, including median, first and third quartiles, and range, for each subjective image quality measure. The reading scores for each image were compared with the Wilcoxon test corrected for multiple comparisons according to the Bonferroni adjustment. Statistical analyses were performed using MedCalc version 18.6 (MedCalc Software, Ostend, Belgium). A *p* value of <0.025 was considered statistically significant.

## 3. Results

### 3.1. Subjective Image Quality

The ratings for artifacts were highest for SSFSE_CS_ (4.15 ± 0.64; median, 4; interquartile range (IQR), 4–5) with a significant difference between both SSFSE_CONV_ (3.75 ± 0.67; median, 4; IQR, 3–4; *p* < 0.001) and mTSE (3.43 ± 0.77; median, 3; IQR, 3–4; *p* < 0.001). mTSE was rated lowest regarding artifacts, with a significant difference compared with SSFSE_CONV_ (*p* = 0.002) (Figure 1, Table 2). 

The sharpness of the organ margin was rated lowest for SSFSE_CONV_ (3.21 ± 0.54; median, 3; IQR, 3–3.25), with a significant difference between mTSE (4.32 ± 0.81; median, 5; IQR, 4–5; *p* < 0.001) and SSFSE_CS_ (4.27 ± 0.58; median, 4; IQR, 4–5; *p* < 0.001). mTSE and SSFSE_CS_ showed no significant difference (*p* = 0.554) (Figure 2, Table 2).

The visibility of small structures, such as the peripheral vessel or duct, was rated highest for SSFSE_CS_ (4.31 ± 0.58; median, 4; IQR, 4–5) with a significant difference compared with SSFSE_CONV_ (3.30 ± 0.59; median, 3; IQR, 3–4; *p* < 0.001) and mTSE (3.82 ± 1.03; median, 4; IQR, 3–5; *p* < 0.001) (Figure 2, Table 2).

We considered overall image quality insufficient in 1 case (1.0%) in SSFSE_CONV_ and 14 cases (13.9%) in mTSE. None of cases were rated insufficient in SSFSE_CS_. The overall image quality was rated highest for SSFSE_CS_ (4.16 ± 0.56; median, 4; IQR, 4–4.25) followed by mTSE (3.68 ± 0.96; median, 4; IQR, 3–4; *p* < 0.001) and SSFSE_CONV_ (3.35 ± 0.50; median, 3; IQR, 3–4; *p* < 0.001) with a statistical significance. SSFSE_CONV_ was rated lowest, with a significant difference compared with mTSE (*p* = 0.003) (Table 2).

### 3.2. Lesion Assessment

The conspicuity of liver lesions was rated highest for SSFSE_CS_ (4.30 ± 0.82; median, 5; IQR, 4–5), with a significant difference between SSFSE_CONV_ (3.35 ± 0.73; median, 3; IQR, 3–4; *p* < 0.001) and mTSE (3.52 ± 1.28; median, 4; IQR, 3–5; *p* < 0.001). SSFSE_CONV_ and mTSE did not significantly differ (*p* = 0.404) (Figure 3) (Table 2). The number of cases with insufficient liver lesion conspicuity (rated 1 or 2) were 5 (9.3%) in SSFSE_CONV_, 11 (20.4%) in mTSE, and 0 in SSFSE_CS_.

The conspicuity of pancreatic cystic lesions was rated highest for SSFSE_CS_ (4.86 ± 0.36; median, 5; IQR, 5–5) with a significant difference between SSFSE_CONV_ (4.00 ± 0.96; median, 4; IQR, 4–5; *p* = 0.008) and mTSE (2.00 ± 0.96; median, 2; IQR, 1–3; *p* < 0.001). mTSE was rated lowest with a significant difference compared with SSFSE_CONV_ (*p* < 0.001) (Figure 4) (Table 2). The numbers of cases with insufficient pancreatic cystic lesion conspicuity (rated 1 or 2) were two (12.5%) in SSFSE_CONV_, eight (50%) in mTSE, and zero in SSFSE_CS_.

## 4. Discussion

In this study, we proved that when the CS technique was applied, subjective image quality was significantly improved compared with that of SSFSE_CONV_. In addition, it was proven to be even better than mTSE, except in organ sharpness. Regarding the conspicuity of liver lesions, SSFSE_CS_ was rated significantly better than the other two sequences, while SSFSE_CONV_ and mTSE did not significantly differ. The conspicuity of pancreatic cystic lesions was rated almost perfect on SSFSE_CS_, while mTSE was rated the lowest among them. 

The CS technique allows accelerated MR signal acquisition by acquiring less data through undersampling of the k-space [12,13]. This method exploits the data sparsity of MR images and uses a nonlinear optimization method to reconstruct the undersampled data [14]. In this regard, a CS technique can be used to shorten the acquisition time while maintaining the image quality, or it can improve image quality while maintaining the acquisition time. Our study demonstrated that when the CS technique was applied in a single BH SSFSE, the image quality was significantly better than that produced with SSFSE_CONV,_ and was even better than that of mTSE. We carefully think that the combination of single BH acquisition and iterative reconstruction of the CS technique might have attributed to improving the image quality of SSFSE_CS_. Even though SSFSE_CS_ requires a reconstruction time of about 30–40 s after its signal acquisition, this is much shorter than required for SSFSE_CONV_ or mTSE, which use three to four BHs, or even RT, which takes at least 2–3 min. In addition, because image reconstruction proceeds as a background process, it does not alter the overall MR workflow.

Several artifacts, including a global ringing mimicking a motion artifact and blurring of fine details, are known to be associated with the CS technique [15,16,17]. Several studies have reported slight blurring or an artificial, blotchy appearance on CS -reconstructed images [15,17]. This may be related to the denoising effect of CS reconstruction, and the artificial, blotchy appearance might also be related to the nature of iterative reconstruction, which was also reported on CT images using iterative reconstruction for reducing radiation dose [18]. In our study, some cases showed a mild blotchy appearance on SSFSE_CS_, but it did not affect the ratings for subjective image quality. None of the cases showed global ringing artifacts.

Regarding the conspicuity of liver lesions, SSFSE_CS_ was significantly better than both SSFSE_CONV_ and mTSE. In cases of well-acquired mTSE, lesion conspicuity as well as overall image quality were better than those of SSFSE_CONV_ in most cases. However, if there was any kind of artifact on mTSE, the image quality was regarded as similar to or even poorer than that of SSFSE_CONV_. With SSFSE_CS_, most cases showed above-average quality even in cases with poor image quality on SSFSE_CONV_ and mTSE. This might be related to the fact that SSFSE_CS_ was acquired in a single BH (which may have contributed to less motion artifacts) and the iterative reconstruction of the CS technique (which is important for improving image quality).

Due to the nature of single-shot T2WI, T2 high-signal-intensity lesions, such as cyst or hemangioma, are generally well-delineated on SSFSE sequences, which was shown on pancreatic cystic lesions. mTSE, which uses multishot for signal acquisition, is prone to motion artifacts including bowel-peristalsis-related artifacts as well as pulsation artifacts. This must have resulted in poor image quality regarding the visualization of pancreatic cystic lesions on mTSE. An advantage of an SSFSE sequence is that it reduces the problem of motion sensitivity by collecting all the data needed for reconstructing a single slice in a fraction of a second. However, the downside is long echo trains that cause the decaying of transversal magnetization, which results in T2 blurring and a low signal yield. As a result, lesions with midrange T2 signal intensity (e.g., macromolecules with fewer protons) can be missed with the SSFSE sequence, which limits its diagnostic accuracy in detecting and characterizing focal liver lesions [19]. This is true for SSFSE_CONV_, but when the CS technique was applied, the lesion conspicuity of the liver and pancreas, as well as the overall image quality, significantly improved. Because iterative reconstruction of the CS technique has potential to reduce image noise and correct aliasing artifacts, we speculate that both lesion conspicuity and overall image quality were improved on SSFSE_CS_.

In a recent study by Chen et al., a non-Cartesian wave-encoded variable-density SSFSE sequence was developed to accelerate the imaging time, and a data-driven, deep-learning based reconstruction was developed to reduce the long computational time required for self-calibration, parallel imaging, and CS reconstruction in non-Cartesian wave-encoded variable-density SSFSE sequences [20,21]. Although image quality was assessed with a phantom experiment, the number of clinical patients was too small and only comparisons between the two SSFSE sequences were performed. In addition, data-driven approaches have great future potential, but require large amounts of data and are still limited in their ability to work with the various scan protocols used in clinical practice.

Most recently, deep learning (DL)-based image reconstruction has been successfully applied to SSFSE sequence and showed comparable or even better image quality compared with SSFSE_CONV_, BLADE, and mTSE [22,23,24]. Compared with the CS technique, one major advantage of DL is that it uses trained regularizers, which are specifically trained to reconstruct a specific type of image. These trained regularizers are considered more efficient than regularizers used in the CS technique. Future studies comparing DL and CS techniques applied to SSFSE are needed in order to reveal the differences between the two techniques using clinical images.

There are several limitations in the current study. First, due to the retrospective nature of the study, there might have been selection bias. Second, although the image review was performed blinded, an experienced reviewer could determine which technique was used. However, there was an extra effort to crop all the images to remove any noticeable pulsation or motion artifacts that were prominent on mTSE images. Third, we did not analyze the BLADE sequence, which is known as T2WI with excellent image quality. Future studies including this sequence are needed to clarify the strengths and weaknesses of SSFSE_CS_. Fourth, we did not evaluate the discrepancy in the lesion detection rate or diagnostic accuracy between the three sequences in detail, as it was beyond the scope of the current study. Fifth, we did not perform quantitative analysis because of the variability in image acquisition and reconstruction techniques among the three T2WIs. Lastly, variable bandwidth and constant refocusing flip angles were used for all SSFSE. It is well-known that changing the bandwidth for SSFSE can lead to nonintuitive effects, and variable refocusing flip angles can be used to more efficiently reduce the SAR and enable a shorter TR, which have potential for shortening the total acquisition time. Future studies with fixed bandwidth as well as utilizing variable refocusing flip angles for comparison of SSFSE are needed.

## 5. Conclusions

SSFSE_CS_ showed superior image quality compared with both SSFSE_CONV_ and mTSE. Based on our study result, SSFSE_CS_ may be used to replace conventional T2-weighted sequences with a much shorter acquisition time. This will help improve the overall workflow of MR imaging as well as increase the availability of MRI to more patients and reduce costs in daily practice.

## Figures and Tables

**Figure 1 diagnostics-12-02164-f001:**
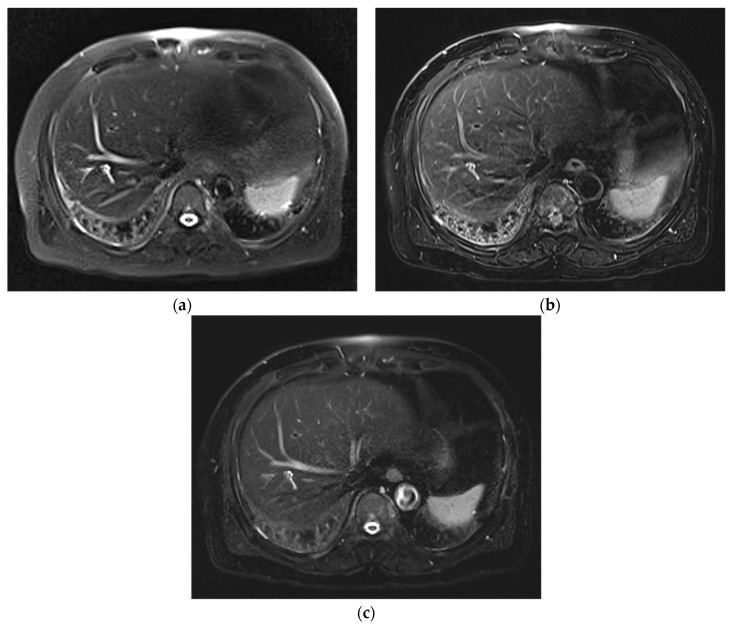
Comparison of image artifacts in three T2WIs acquired on Skyra. SSFSE_CONV_ (**a**) was rated 3, mTSE (**b**) was rated 2, and SSFSE_CS_ (**c**) was rated 4. On SSFSE_CONV_ and mTSE, cardiac motion artifacts, as well as some respiratory motion artifacts, are present.

**Figure 2 diagnostics-12-02164-f002:**
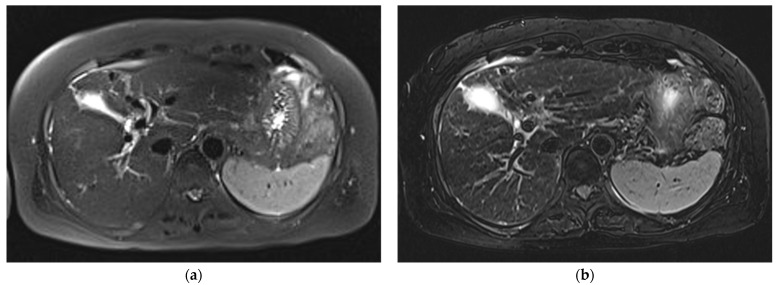
Comparison of sharpness of organ margin in three T2WIs acquired on Skyra. SSFSE_CONV_ (**a**) was rated 2, mTSE (**b**) was rated 5, and SSFSE_CS_ (**c**) was rated 4. The visibility of peripheral vessel or duct was rated highest for mTSE (5), followed by SSFSE_CS_ (4) and SSFSE_CONV_ (2).

**Figure 3 diagnostics-12-02164-f003:**
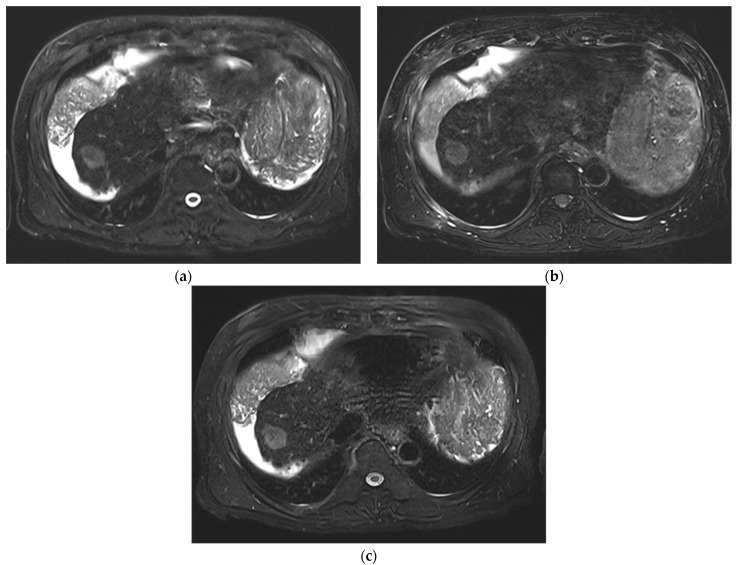
Comparison of conspicuity of the main liver lesion (HCC) in three T2WIs acquired on Vida. SSFSE_CONV_ (**a**) and mTSE (**b**) were rated 3, while SSFSE_CS_ (**c**) was rated 5.

**Figure 4 diagnostics-12-02164-f004:**
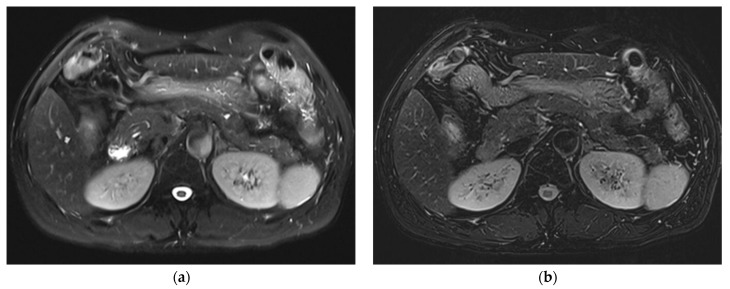
Comparison of conspicuity of main pancreatic cystic lesion (IPMN) in three T2WIs acquired on Vida. SSFSE_CONV_ (**a**) was rated 4, mTSE (**b**) was rated 1, and SSFSE_CS_ (**c**) was rated 5.

**Table 1 diagnostics-12-02164-t001:** Acquisition parameters of three T2-weighted images.

Parameter	Skyra (*n* = 52)			Vida (*n* = 49)		
Sequence	SSFSE_CONV_	mTSE	SSFSE_CS_	SSFSE_CONV_	mTSE	SSFSE_CS_
TR/TE (ms)	732/100	2040/101	500/99	1000/99	1700/99	561/96
FA, degree	106	134	118	154	140	135
FOV (mm)	380 × 380	380 × 380	380 × 309	380 × 309	380 × 380	380 × 309
Matrix	256 × 256	384 × 307	384 × 253	384 × 250	384 × 307	384 × 253
ST (mm)	5	5	5	5	5	5
Number of slices	42	42	32	42	42	32
Motion management	3 BH	4 BH	1 BH	3 BH	4 BH	1 BH
ETL	128	22	84	125	17	84
BW (Hz)	977	1085	372	372	303	685
Acceleration factor	GRAPPA = 2	GRAPPA = 2	CS = 3	GRAPPA = 2	GRAPPA = 2	CS = 3
Scan time (minute)	0:58	1:48	0:18	1:02	1:48	0:18

SSFSE_CONV_, conventional SSFSE; mTSE, multishot turbo spin-echo; SSFSE_CS_, compressed sensing accelerated SSFSE; TR/TE, repetition time/echo time; FA, flip angle; FOV, field of view; ST, slice thickness; BH, breath-hold; ETL, echo train length; BW, bandwidth; GRAPPA, generalized autocalibrating partially parallel acquisition; CS, compressed sensing.

**Table 2 diagnostics-12-02164-t002:** Results of subjective image analysis using five-point evaluation scale system for each MRI sequence.

	Artifact	Organ Sharpness	Visibility of Small Structures	Overall Image Quality	Conspicuity of Liver Lesion	Conspicuity of Pancreatic Cystic Lesion
SSFSE_CONV_	3.75 ± 0.67	3.21 ± 0.54	3.30 ± 0.59	3.35 ± 0.50	3.35 ± 0.73	4.00 ± 0.96
TSE	3.43 ± 0.77	4.32 ± 0.81	3.82 ± 1.03	3.68 ± 0.96	3.52 ± 1.28	2.00 ± 0.96
SSFSE_CS_	4.15 ± 0.64	4.27 ± 0.58	4.31 ± 0.58	4.16 ± 0.56	4.30 ± 0.82	4.86 ± 0.36
*p* value	0.002 *, <0.001 ^†‡^	<0.001 *^†^, 0.554 ^‡^	<0.001 *^†‡^	0.003 *, <0.001 ^†‡^	0.404 *, <0.001 ^†‡^	<0.001 *^‡^, 0.008 ^†^

Values are mean ± standard deviation. SSFSE_CONV_, conventional SSFSE; TSE, turbo spin-echo; SSFSE_CS_, SSFSE acquired with compressed-sensing technique. *p* values are for comparison of * SSFSE_CONV_ and TSE, ^†^ SSFSE_CONV_ and SSFSE_CS_, and ^‡^ TSE and SSFSE_CS_.

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
