# Peer review of "Improved Single Breath-Hold SSFSE Sequence for Liver MRI Based on Compressed Sensing: Evaluation of Image Quality Compared with Conventional T2-Weighted Sequences"

_diagnostics, 2022, doi:10.3390/diagnostics12092164_

Round 1

Reviewer 1 Report

Dear Authors,

In this study image quality of HASTE sequence, with and without compressed sensing and turbo spin echo were evaluated in 101 patients for liver and pancreas.  The authors showed the interest of compressed sensing apply to HASTE sequence, HASTEcs image quality was significantly better than HASTEconv and mTSE image quality for artifacts, small structure visibility, overall image quality, and conspicuity of main lesions.

However, there are a number of issues to be addressed:

1.      Abstract. Please indicate the acquisition time in seconds, this is the physical units conventionally used.

2.      Method – Statistical analysis. The manuscript would benefit from a statistical analysis of the intra-class correlation coefficient to assess consistency in reader scores.

3.      Results. The methods indicate the used of two MR systems with quite different MR parameter. Then, it will be interesting to apply a separate analysis relatively to the MR system used to investigate its effect on image quality and eliminate the bias of the MR system.

4.      Figures. Please modify the figure they are quite large and they are separate in two pages.

5.      Discussion – p8, l215-216. ‘’ Conventional HASTE could also be acquired in a single BH, although the image quality would be very poor with a limited scan range.’’ It will be interesting to have a sentence explaining why the image quality is better with HASTEcs. Which parameter has the main impact a single breath-hold or the matrix resolution…

6.      Discussion – p8, 252-254. ‘’This is true for HASTECONV, but when the CS technique was applied, the lesion conspicuity of the liver and pancreas, as well as the overall image quality, significantly improved.’’ Could the authors explain why results are improved when compressed sensing is applied?

7.      Discussion. A publication from Shanbhogue et al. in European Radiology used an accelerated single-shot T2-weighted fat-suppressed MRI of the liver with deep learning–based image reconstruction, it will be interesting to discuss the results obtained here with those obtained by Shanbhogue et al.

Author Response

Reviewer: #1

In this study image quality of HASTE sequence, with and without compressed sensing and turbo spin echo were evaluated in 101 patients for liver and pancreas.  The authors showed the interest of compressed sensing apply to HASTE sequence, HASTEcs image quality was significantly better than HASTEconv and mTSE image quality for artifacts, small structure visibility, overall image quality, and conspicuity of main lesions.

However, there are a number of issues to be addressed:

1-1. Abstract. Please indicate the acquisition time in seconds, this is the physical units conventionally used.

Answer) Thank you for your comment. We’ve indicated the acquisition time in seconds according to your suggestion.

1-2. Method – Statistical analysis. The manuscript would benefit from a statistical analysis of the intra-class correlation coefficient to assess consistency in reader scores.

Answer) We appreciate your comment. However, all three T2WI assessments were done by a consensus reading of two radiologists.

1-3. Results. The methods indicate the used of two MR systems with quite different MR parameter. Then, it will be interesting to apply a separate analysis relatively to the MR system used to investigate its effect on image quality and eliminate the bias of the MR system.

Answer) Thank you for your comment. Since the two MR systems used were from the same vendor (Siemens Healthineers), there were no significant difference in its subjective image quality. In addition, we carefully assume that direct comparison of two MR system with statistical analysis is beyond the scope of our current study. Instead, we added the number of patients of each MR machine used in Table 1 and mentioned which MR machine was used in all 4 figures.

1-4. Figures. Please modify the figure they are quite large and they are separate in two pages.

Answer) Thank you for your comment. We will discuss this issue with the editorial office.

1-5. Discussion – p8, l215-216. ‘’ Conventional HASTE could also be acquired in a single BH, although the image quality would be very poor with a limited scan range.’’ It will be interesting to have a sentence explaining why the image quality is better with HASTEcs. Which parameter has the main impact a single breath-hold or the matrix resolution…

Answer) Thank you for your comment. We’ve added the following sentence in the discussion section: “We carefully assume that combination of single BH acquisition and iterative reconstruction of the CS technique might have attributed to improving the image quality of HASTECS

1-6. Discussion – p8, 252-254. ‘’This is true for HASTECONV, but when the CS technique was applied, the lesion conspicuity of the liver and pancreas, as well as the overall image quality, significantly improved.’’ Could the authors explain why results are improved when compressed sensing is applied?

Answer) We appreciate your comment. We’ve added the following sentence in the discussion section: “Since iterative reconstruction of the CS technique has potential to reduce image noise and correct aliasing artifacts, we speculate that both lesion conspicuity and overall image quality was improved on HASTECS

1-7. Discussion. A publication from Shanbhogue et al. in European Radiology used an accelerated single-shot T2-weighted fat-suppressed MRI of the liver with deep learning–based image reconstruction, it will be interesting to discuss the results obtained here with those obtained by Shanbhogue et al.

Answer) Thank you for your comment. We’ve added a paragraph in the discussion section mentioning study by Shanbhogue et al and 2 more studies about deep learning-based image reconstruction.

Reviewer 2 Report

This manuscript describes a SSFSE sequence utilizing variable density k-space sampling, with compressed sensing reconstruction utilized to make up for the missing data. The authors compare this sequence to a conventional SSFSE sequence and a fast spin echo (FSE) sequence using a two-reader Likert scale type evaluation, and show their new sequence outperforms the comparison sequences in several assessed parameters. The evaluation methods are commonly used in these types of sequence comparison studies, and there are no major methodological flaws to the study. 

The major criticisms of the study are due to

1) Inadequate explanation of the sequence parameters for the sequences compared. Because of this, it’s not at all clear why one sequence worked better than another. A major concern is that the comparison sequence (termed HASTEconv by the authors) may represent a fairly old implementation of SSFSE and may not incorporate improvements that have manifested over the last decade such as variable refocusing flip angles, higher acceleration factors, high-density coil arrays, etc. It is critical in these sort of studies that as few parameters as possible are being changed between the “old” sequence and the “new” sequence.

2) No referencing and discussion of prior work in the field that has previously implemented variable density techniques with compressed sensing reconstruction in SSFSE (HASTE) sequences.

General Comments: 

* HASTE is a vendor specific term. SSFSE is to my understanding the more commonly used term in the literature and more generic term. I would suggest calling the sequences “SSFSEcs” and “SSFSEconv”, rather than using the term “HASTE”

* I believe fast spin echo is the more commonly used term, rather than turbo spin echo.

Introduction

* The manuscript would benefit from a discussion on the impact of T2-decay related blurring, as this is one mechanism whereby the author’s technique may improve on image quality versus the conventional SSFSE that they are comparing to.

Methods:

* The methods section does not include nearly enough information to understand what the authors are comparing.

1) Is the SSFSE technique the authors are using actually Half-Fourier (the term “HASTE” implies half-Fourier, which is non-ideal)? What percentage of k-space is actually covered.

2) How is the missing portion of k-space interpolated if it’s partial Fourier, is it homodyne reconstruction?

3) What were the refocusing echo train flip angles utilized for their sequences? Were these fixed or variable…. the choice of the flip angles has huge impacts on the degree of T2-decay related blurring as well as SAR constraints (see Loening 2015, Busse 2006, etc.)

4) Why were the SSFSEconv (termed by the authors HASTEconv) sequences such a long acquisition? Since they imaged at 3T, the long acquisition time may have been due to SAR constraints from using a fixed refocusing echo train (which is also non-optimal for image quality due to T2-decay related blurring). Was this the stock vendor sequence? If this is the vendor sequence, which version of the vendor’s software is being used?

5) What were the acceleration factors utilized for each sequence? 

6) ETL’s listed in Table 1 are likely inaccurate. The authors state the effective acceleration for SSFSEcs (HASTEcs) with the VD k-space samplling pattern was 3, and they acquired calibration data independent of their acquisition, so ETL should be at most 253/3. For  SSFEconv (HASTEconv), these acquisitions are generally accelerated (most typically a factor of 2), and only use partial k-space acquisition in order to hit their stated echo time, so I would expect something closer to ~70-110 for the ETL.

7) More more info is needed on the VD k-space trajectory. Was it Cartesian? How was the density of k-space acquisition determined? Can a figure be provided showing the sampling pattern of k-space?

8) How many coil elements were utilized? 

9) What was the phase encode direction?

10) Why was bandwidth so different between the different sequences? Why was the bandwidth so difference for the same sequence (HASTEconv) between the 2 magnets utilized?

Findings

* p-values should be corrected for multi-comparisons. Given the high level of significance, most of the significantly significant differences will likely remain

* an inter-observer assessment should be performed.

Discussion

“This is the first study to apply a CS technique to motion robust single-shot T2 TSE,  HASTE.” - The authors’ manuscript is not the first study. The authors need to acknowledge prior work in the field, and compare their work to this prior work. Specific work that needs to be acknowledged include:

* Self-Calibrating Wave-Encoded Variable-Density Single-Shot Fast Spin Echo Imaging.

Chen F, Taviani V, Tamir JI, Cheng JY, Zhang T, Song Q, Hargreaves BA, Pauly JM, Vasanawala SS.

J Magn Reson Imaging. 2018 Apr;47(4):954-966. doi: 10.1002/jmri.25853. Epub 2017 Sep 14.

PMID: 28906567

* Data-driven self-calibration and reconstruction for non-cartesian wave-encoded single-shot fast spin echo using deep learning.

Chen F, Cheng JY, Taviani V, Sheth VR, Brunsing RL, Pauly JM, Vasanawala SS.

J Magn Reson Imaging. 2020 Mar;51(3):841-853. doi: 10.1002/jmri.26871. Epub 2019 Jul 19.

PMID: 31322799

There is also extensive work that has been published as conference abstracts. In the MRI world it’s fairly common for research to only be presented in abstract form at the ISMRM meeting, and not always make it into a journal manuscript form.

* Variable Density Compressed Sensing Single Shot Fast Spin Echo

Valentina Taviani, Daniel V. Litwiller, Jonathan I. Tamir, Andreas M. Loening, Brian A. Hargreaves, and Shreyas S. Vasanawala

https://cds.ismrm.org/protected/16MProceedings/PDFfiles/0618.html

* Improved Speed and Image Quality for Imaging of Liver Lesions with Auto-calibrated Wave Encoded Variable Density Single-Shot Fast Spin Echo.

Jamil Shaikh, Feiyu S. Chen, Valentina S. Taviani, Kim Nhien Vu, and Shreyas S. Vasanawala

https://cds.ismrm.org/protected/18MProceedings/PDFfiles/0998.html

“We proved that when the CS technique was applied, subjective image quality was significantly improved compared with HASTECONV” - Although the authors results show this, not enough technical details are given on how their new sequence works to explain why their sequence may be better.

“Conventional HASTE could also be acquired in a single BH, although the image quality would be very poor with a limited scan range.” - Conventional SSFSE is commonly acquired in a single breath hold without a limited scan range, with high image quality. 

Author Response

Reviewer: #2

This manuscript describes a SSFSE sequence utilizing variable density k-space sampling, with compressed sensing reconstruction utilized to make up for the missing data. The authors compare this sequence to a conventional SSFSE sequence and a fast spin echo (FSE) sequence using a two-reader Likert scale type evaluation, and show their new sequence outperforms the comparison sequences in several assessed parameters. The evaluation methods are commonly used in these types of sequence comparison studies, and there are no major methodological flaws to the study.

The major criticisms of the study are due to

1) Inadequate explanation of the sequence parameters for the sequences compared. Because of this, it’s not at all clear why one sequence worked better than another. A major concern is that the comparison sequence (termed HASTEconv by the authors) may represent a fairly old implementation of SSFSE and may not incorporate improvements that have manifested over the last decade such as variable refocusing flip angles, higher acceleration factors, high-density coil arrays, etc. It is critical in these sort of studies that as few parameters as possible are being changed between the “old” sequence and the “new” sequence.

2) No referencing and discussion of prior work in the field that has previously implemented variable density techniques with compressed sensing reconstruction in SSFSE (HASTE) sequences.

General Comments: 

2-1. * HASTE is a vendor specific term. SSFSE is to my understanding the more commonly used term in the literature and more generic term. I would suggest calling the sequences “SSFSEcs” and “SSFSEconv”, rather than using the term “HASTE”

* I believe fast spin echo is the more commonly used term, rather than turbo spin echo.

Answer) Thank you for your comment. We’ve changed the term HASTE to SSFSE in all of our manuscript.

Introduction

2-2. The manuscript would benefit from a discussion on the impact of T2-decay related blurring, as this is one mechanism whereby the author’s technique may improve on image quality versus the conventional SSFSE that they are comparing to.

Answer) We appreciate your comment. We have prepared a paragraph regarding the impact of T2-decay related blurring in the discussion section.

Methods: * The methods section does not include nearly enough information to understand what the authors are comparing.

2-3.  Is the SSFSE technique the authors are using actually Half-Fourier (the term “HASTE” implies half-Fourier, which is non-ideal)? What percentage of k-space is actually covered.

Answer) Thank you for your comment. CS reconstruction does not make any assumptions about unmeasured segments in k-space. The acceleration factor 3 is used for this study, it covers 33.33% of k-space. We’ve added an information about acceleration factor in Table 1.

2-4. How is the missing portion of k-space interpolated if it’s partial Fourier, is it homodyne reconstruction?

Answer) Yes, additional homodyne filter is applied after CS recon of partial Fourier acquisition.

2-5. What were the refocusing echo train flip angles utilized for their sequences? Were these fixed or variable…. the choice of the flip angles has huge impacts on the degree of T2-decay related blurring as well as SAR constraints (see Loening 2015, Busse 2006, etc.)

Answer) The flip angle was taken with a constant FA at the level where SAR is not applied in both of conventional SSFSE and CS-SSFSE as the conventional SSFSE sequence does not have a variable FA option.

2-6.  Why were the SSFSEconv (termed by the authors HASTEconv) sequences such a long acquisition? Since they imaged at 3T, the long acquisition time may have been due to SAR constraints from using a fixed refocusing echo train (which is also non-optimal for image quality due to T2-decay related blurring). Was this the stock vendor sequence? If this is the vendor sequence, which version of the vendor’s software is being used?

Answer) Yes, the conventional SSFSE used in this study was the stock vendor sequence, and Syngo MR XA31 software was used. The reason why the total acquisition time is longer in conventional SSFSE is because the TR was longer with GRAPPA 2 and the number of slices was slightly higher, so three BHs were required.

2-7. What were the acceleration factors utilized for each sequence? 

Answer) GRAPPA 2 in conventional SSFSE was used, and CS factor 3 was used in CS-SSFSE. We’ve added an information about acceleration factor in Table 1.

2-8. ETL’s listed in Table 1 are likely inaccurate. The authors state the effective acceleration for SSFSEcs (HASTEcs) with the VD k-space samplling pattern was 3, and they acquired calibration data independent of their acquisition, so ETL should be at most 253/3. For  SSFEconv (HASTEconv), these acquisitions are generally accelerated (most typically a factor of 2), and only use partial k-space acquisition in order to hit their stated echo time, so I would expect something closer to ~70-110 for the ETL.

Answer) Thank you for your comment. Yes, you are right, the ETL listed in the table does not reflect GRAPPA and CS acceleration. So we’ve added an information about acceleration factor in Table 1.

2-9. More more info is needed on the VD k-space trajectory. Was it Cartesian? How was the density of k-space acquisition determined? Can a figure be provided showing the sampling pattern of k-space?

Answer) The VD sampling performs a variable density reordering in phase encoding direction of Cartesian trajectory with a total acceleration and the increase of the acceleration from k-space center to periphery specified by acceleration increment. We’ve rephrased ‘2.3. Compressed Sensing Accelerated SSFSE Sequence’ as appropriate.

2-10. How many coil elements were utilized? 

Answer) Thank you for your comment. The combination of 18 channel (Skyra) and 30 channel (Vida) flexible anterior body coil and 32 channel spine matrix coil were utilized.

2-11. What was the phase encode direction?

Answer) AP direction at Axial plane was used.

2-12. Why was bandwidth so different between the different sequences? Why was the bandwidth so difference for the same sequence (HASTEconv) between the 2 magnets utilized?

Answer) Thank you for your comment, and we agree that this is another limitation. It was set to maintain the SNR a little more because it was possible to scan under the same conditions as 3BH without having to increase the BW. We didn't match all the parameters because the two sequences are completely different from acquisition to recon. So, we set the protocol flexibly to improve the image quality within a sequence.

Findings

2-13.  p-values should be corrected for multi-comparisons. Given the high level of significance, most of the significantly significant differences will likely remain

Answer) We appreciate your comment. We’ve already used Friedman test (which is a test used for comparing three or more matched groups) for statistical analysis and confirmed there were statistical significance. However, the results obtained with Friedman test did not have p values of each groups comparison (e.g. conventional SSFSE vs mTSE or conventional SSFSE vs CS-SSFSE) so we decided to use Wilcoxon signed-rank test.

2-14. an inter-observer assessment should be performed.

Answer) We appreciate your comment. However, all three T2WI assessments were done by a consensus reading of two radiologists.

2-15. This is the first study to apply a CS technique to motion robust single-shot T2 TSE,  HASTE.” - The authors’ manuscript is not the first study. The authors need to acknowledge prior work in the field, and compare their work to this prior work. Specific work that needs to be acknowledged include:

* Self-Calibrating Wave-Encoded Variable-Density Single-Shot Fast Spin Echo Imaging.

Chen F, Taviani V, Tamir JI, Cheng JY, Zhang T, Song Q, Hargreaves BA, Pauly JM, Vasanawala SS.

J Magn Reson Imaging. 2018 Apr;47(4):954-966. doi: 10.1002/jmri.25853. Epub 2017 Sep 14.

PMID: 28906567

* Data-driven self-calibration and reconstruction for non-cartesian wave-encoded single-shot fast spin echo using deep learning.

Chen F, Cheng JY, Taviani V, Sheth VR, Brunsing RL, Pauly JM, Vasanawala SS.

J Magn Reson Imaging. 2020 Mar;51(3):841-853. doi: 10.1002/jmri.26871. Epub 2019 Jul 19.

PMID: 31322799

There is also extensive work that has been published as conference abstracts. In the MRI world it’s fairly common for research to only be presented in abstract form at the ISMRM meeting, and not always make it into a journal manuscript form.

* Variable Density Compressed Sensing Single Shot Fast Spin Echo

Valentina Taviani, Daniel V. Litwiller, Jonathan I. Tamir, Andreas M. Loening, Brian A. Hargreaves, and Shreyas S. Vasanawala

https://cds.ismrm.org/protected/16MProceedings/PDFfiles/0618.html

* Improved Speed and Image Quality for Imaging of Liver Lesions with Auto-calibrated Wave Encoded Variable Density Single-Shot Fast Spin Echo.

Jamil Shaikh, Feiyu S. Chen, Valentina S. Taviani, Kim Nhien Vu, and Shreyas S. Vasanawala

https://cds.ismrm.org/protected/18MProceedings/PDFfiles/0998.html

Answer) Thank you for your comment. We’ve added some of the references in our manuscript and revised the discussion section as appropriate.

2-16. “We proved that when the CS technique was applied, subjective image quality was significantly improved compared with HASTECONV” - Although the authors results show this, not enough technical details are given on how their new sequence works to explain why their sequence may be better.

Answer) We appreciate your comment. We’ve worked hard to make substantial changes regarding the technical details according to your suggestions.

2-17. “Conventional HASTE could also be acquired in a single BH, although the image quality would be very poor with a limited scan range.” - Conventional SSFSE is commonly acquired in a single breath hold without a limited scan range, with high image quality. 

Answer) Thank you for your comment. Conventional SSFSE requires a high GRAPPA factor to acquire all slices in a single BH. However, higher GRAPPA factor produces aliasing artifact and low SNR. CS can provide more acceleration while allowing de-nosing and reduces the ETL length, resulting in shorter TRs. Neither sequence used VFA in this study, but VFA may help to reduce SAR, enabling shorter TR.

Reviewer 3 Report

The present study found that HASTECS outperformed HASTECONV and mTSE sequences in assessing the image quality of T2WI in liver MRI. HASTECS may serve as an alternative to conventional T2-weighted sequences with a shorter scan time. It is a good study and well-written, but a few concerns need to be resolved for the current version of the manuscript.

The authors used two MR systems, how about the differences in image quality between them using different imaging parameters?

Since the authors only qualitatively evaluated the image quality of the three sequences, they should perform quantitative metrics as well, such as signal-to-noise ration and contrast-to-noise ratio. In addition, the reproducibility of them should also be evaluated.

Overall, the references are old, please add or update several latest ones.

Author Response

Reviewer: #3

The present study found that HASTECS outperformed HASTECONV and mTSE sequences in assessing the image quality of T2WI in liver MRI. HASTECS may serve as an alternative to conventional T2-weighted sequences with a shorter scan time. It is a good study and well-written, but a few concerns need to be resolved for the current version of the manuscript.

3-1. The authors used two MR systems, how about the differences in image quality between them using different imaging parameters?

Answer) Thank you for your comment. Since the two MR systems used were from the same vendor (Siemens Healthineers), there were no significant difference in its subjective image quality. In addition, we carefully assume that direct comparison of two MR system with statistical analysis is beyond the scope of our current study. Instead, we added the number of patients of each MR machine used in Table 1 and mentioned which MR machine was used in all 4 figures.

3-2. Since the authors only qualitatively evaluated the image quality of the three sequences, they should perform quantitative metrics as well, such as signal-to-noise ration and contrast-to-noise ratio. In addition, the reproducibility of them should also be evaluated.

Answer) Thank you for your comment. Since we analyzed three different T2WIs acquired and reconstructed with various techniques, we assumed that it would be inappropriate to compare SNR and/or CNR directly. We’ve added the lack of quantitative analysis as a limitation in the discussion section.

3-3. Overall, the references are old, please add or update several latest ones.

Answer) We appreciate your comment. We’ve added some of the recent studies, especially deep learning-based reconstruction in the discussion section, as appropriate.

Reviewer 4 Report

H. K., et al., and coworkers reported a review article “Improved Single Breath-Hold HASTE Sequence for Liver MRI Based on Compressed-Sensing: Evaluation of Image Quality Compared to Conventional T2-Weighted Sequences”. Here in this article, the authors compared the image quality of compressed-sensing accelerated half-Fourier single-shot turbo spin echo (HASTECS) sequences acquired within a single breath-hold in comparison to conventional HASTE (HASTECONV) and multishot TSE (mTSE) in ~101 patients who underwent Liver MRI retrospectively. Both the Descriptive statistics and the Wilcoxon signed-rank test confirmed that the HASTECS was significantly better than HASTECONV and mTSE for artifacts, small structure visibility, overall image quality, and conspicuity of main lesions (p < 0.005), however, organ sharpness, mTSE and HASTECS (p = 0.554) and Conspicuity of liver lesion did not differ significantly. All taken together, the current study will help improve the overall workflows of MR imaging as well as increase the availability of MRI to more patients and reduce costs in daily practice.

Overall, the MS was well organized and discussed in a good manner. Therefore, I recommend the editor accept the manuscript.

Author Response

Reviewer: #4

H.K., et al., and coworkers reported a review article “Improved Single Breath-Hold HASTE Sequence for Liver MRI Based on Compressed-Sensing: Evaluation of Image Quality Compared to Conventional T2-Weighted Sequences”. Here in this article, the authors compared the image quality of compressed-sensing accelerated half-Fourier single-shot turbo spin echo (HASTECS) sequences acquired within a single breath-hold in comparison to conventional HASTE (HASTECONV) and multishot TSE (mTSE) in ~101 patients who underwent Liver MRI retrospectively. Both the Descriptive statistics and the Wilcoxon signed-rank test confirmed that the HASTECS was significantly better than HASTECONV and mTSE for artifacts, small structure visibility, overall image quality, and conspicuity of main lesions (p < 0.005), however, organ sharpness, mTSE and HASTECS (p = 0.554) and Conspicuity of liver lesion did not differ significantly. All taken together, the current study will help improve the overall workflows of MR imaging as well as increase the availability of MRI to more patients and reduce costs in daily practice.

Overall, the MS was well organized and discussed in a good manner. Therefore, I recommend the editor accept the manuscript.

Round 2

Reviewer 1 Report

Thanks for the authors' response to my concerns

Author Response

Thank you very much.

Reviewer 2 Report

This manuscript describes a SSFSE sequence utilizing variable density k-space sampling, with compressed sensing reconstruction utilized to make up for the missing data. The authors compare this sequence to a conventional SSFSE sequence and a fast spin echo (FSE) sequence using a two-reader Likert scale type evaluation, and show their new sequence outperforms the comparison sequences in several assessed parameters. The evaluation methods are commonly used in these types of sequence comparison studies.

There are several minor criticisms related to insufficient details regarding the sequences utilized that remain to be addressed. The only major remaining criticism (discussed below), is that different bandwidths were utilized between the SSFSEconv and the SSFSEcs sequences.

Comments/Suggestions:

2-3 and 2-4) If the SSFSE acquisition is partial Fourier, which percentage of k-space is actually acquired? This was not answered by the authors in their response, and needs to be stated in the methods. Additionally, the use of a homodyne filter to synthesize the missing k-space data needs to be stated in the methods.

2-5) What were the refocusing flip angles? I understand from the author’s responses that it was a fixed flip angle, but it is still not stated in the text that the flip angles were constant and what flip angles were used? Also, was the SSFSEcs a fixed flip angle, and was it the same as SSFSEconv?

2-6) The number of breath holds and acquisition time cannot be compared between the 2 sequences (SSFESconv and SSFSEcs) as the number of slices was not kept constant. The authors should instead state the acquisition time per slice since they did not keep the number of slices constant, and this is the “speed-up” they should state in the abstract of the manuscript. Also, the details of the vendor sequence (vendor, software version) needs to be included in the text.

2-8) The authors acknowledge that the echo train lengths are incorrect in Table 1, but have not updated table 1. For instance, the echo train for SSFSEconv should be somewhere between ~70 and 140, depending on the degree of k-space that is directly sampled versus inferred (partial Fourier).  Also, since the calibration data is acquired separately with SSFEcs, the number of extra echos utilized for SSFSEcs should be stated somewhere in the table.

2-9) The authors still do not state what variable density pattern is utilized in their answer, or in the text. This needs to be described in the manuscript.

2-11) The phase encode direction utilized for all sequences needs to be described in the methods (anterior-posterior per the authors’ response). 

2-12) The variable bandwidth between the sequences utilized creates a real problem in comparison between the SSFSE sequences. Changing the bandwidth for SSFSE can lead to non-intuitive effects. While in general lower bandwidths allow for better SNR in MRI, in SSFSE this is not necessarily the case. With SSFSE, higher bandwidths allows acquiring the echo train in a short amount of time and decreasing T2-decay related blurring with potentially limited effect on SNR particularly if the T2-decay times of the tissue being evaluated is short. As the bandwidths were so different between the SSFSE sequences compared, there would have been a huge impact on this in their results. Ideally, the authors would repeat their experiments keeping bandwidth fixed between the SSFSE sequences. At a minimum the inconsistency in bandwidth needs to be listed in the text as a limitation of the study.

The authors also need to double check the bandwidths listed in Table 1, as it’s unclear why the bandwidths would change so much between the Skyra and Vida magnets.

2-13) A set p-value cut-off of <0.05 cannot be used when there are multiple comparisons. The authors need to incorporate a correction (such as Holms-Bonferonn) to their p-value cut-off as they are doing multiple comparisons. This has not yet been addressed by the authors and needs to be included in the manuscript.

2-17) The line ““Conventional SSFSE could also be acquired in a single BH, although the image quality would be very poor with a limited scan range.” Needs to be either removed or rewritten as this is incorrect. Conventional SSFSE is often acquired as a single breath hold with reasonable image quality through a variety of techniques that the authors did not implement. These include a) higher bandwidth technique, b) variable flip angles, c) reducing the resolution in the phase direction, d) higher acceleration factors; for an axial acquisition such as in the case of this study the phase direction can be run right-left (with the arms up if outer-volume suppression techniques are not implemented) to allow better utilization of the coil array geometry and acceleration factors of at least 3 if not higher.

As a limitation of this study, the authors need to state that they did not utilize variable refocussing flip angles with SSFSE (available with some vendor implementations such as GE), and that they’re faster acquisition time was largely due to SAR constraints resulting from the constant refocusing flip angles.

2-18) A new question, was the same interleaving pattern used for SSFSEconv and SSFSEcs? From the text it is unclear if the slice increment of 4 was used for just one, or both of the SSFSE sequences. This needs to be better described in the manuscript.

Author Response

2-3., 2-4. If the SSFSE acquisition is partial Fourier, which percentage of k-space is actually acquired? This was not answered by the authors in their response, and needs to be stated in the methods. Additionally, the use of a homodyne filter to synthesize the missing k-space data needs to be stated in the methods.

Answer) Thank you for your comment. The percentage of partial Fourier was 75% (6/8). We’ve added this information and the use of a homodyne filter in the manuscript.

2-5. What were the refocusing flip angles? I understand from the author’s responses that it was a fixed flip angle, but it is still not stated in the text that the flip angles were constant and what flip angles were used? Also, was the SSFSEcs a fixed flip angle, and was it the same as SSFSEconv?

Answer) We used a constant flip angle for refocusing pulses and the degrees are stated in Table 1.

2-6.  The number of breath holds and acquisition time cannot be compared between the 2 sequences (SSFESconv and SSFSEcs) as the number of slices was not kept constant. The authors should instead state the acquisition time per slice since they did not keep the number of slices constant, and this is the “speed-up” they should state in the abstract of the manuscript. Also, the details of the vendor sequence (vendor, software version) needs to be included in the text.

Answer) The acquisition time per slice was 0.7~1.0 sec in SSFSEconv and 0.5~0.6 sec in SSFSEcs. The used system and software version were as follows: MAGNETOM Skyra, VE11E version; MAGNETOM Vida, VA20A version. We’ve added these informations in the manuscript.

2-8. The authors acknowledge that the echo train lengths are incorrect in Table 1, but have not updated table 1. For instance, the echo train for SSFSEconv should be somewhere between ~70 and 140, depending on the degree of k-space that is directly sampled versus inferred (partial Fourier).  Also, since the calibration data is acquired separately with SSFEcs, the number of extra echos utilized for SSFSEcs should be stated somewhere in the table.

Answer) Thank you for your comment. The echo train lengths for SSFSEconv was 125~128 and 84 for SSFSEcs. And yes, 24 automatic calibration signal (ACS) lines for parallel reconstruction were acquired separately, and we included this information in the ‘2.3. Compressed sensing accelerated SSFSE sequence’ section.

2-9. The authors still do not state what variable density pattern is utilized in their answer, or in the text. This needs to be described in the manuscript.

Answer) The VD sampling performs a variable density reordering in phase encoding direction of Cartesian trajectory with a total acceleration and the increase of the acceleration from k-space center to periphery specified by acceleration increment. We’ve already rephrased ‘2.3. Compressed Sensing Accelerated SSFSE Sequence’ in previous revision.

2-11. The phase encode direction utilized for all sequences needs to be described in the methods (anterior-posterior per the authors’ response).

Answer) AP direction at Axial plane was used.

2-12. The variable bandwidth between the sequences utilized creates a real problem in comparison between the SSFSE sequences. Changing the bandwidth for SSFSE can lead to non-intuitive effects. While in general lower bandwidths allow for better SNR in MRI, in SSFSE this is not necessarily the case. With SSFSE, higher bandwidths allows acquiring the echo train in a short amount of time and decreasing T2-decay related blurring with potentially limited effect on SNR particularly if the T2-decay times of the tissue being evaluated is short. As the bandwidths were so different between the SSFSE sequences compared, there would have been a huge impact on this in their results. Ideally, the authors would repeat their experiments keeping bandwidth fixed between the SSFSE sequences. At a minimum the inconsistency in bandwidth needs to be listed in the text as a limitation of the study.

The authors also need to double check the bandwidths listed in Table 1, as it’s unclear why the bandwidths would change so much between the Skyra and Vida magnets.

Answer) Thank you for your comment, and again we agree that this is another limitation. We added this as another limitation of our study.

Findings

2-13.  A set p-value cut-off of <0.05 cannot be used when there are multiple comparisons. The authors need to incorporate a correction (such as Holms-Bonferonn) to their p-value cut-off as they are doing multiple comparisons. This has not yet been addressed by the authors and needs to be included in the manuscript.

Answer) Thank you for your comment. We’ve rephrased the statistical analysis section as appropriate.

2-17. The line ““Conventional SSFSE could also be acquired in a single BH, although the image quality would be very poor with a limited scan range.” Needs to be either removed or rewritten as this is incorrect. Conventional SSFSE is often acquired as a single breath hold with reasonable image quality through a variety of techniques that the authors did not implement. These include a) higher bandwidth technique, b) variable flip angles, c) reducing the resolution in the phase direction, d) higher acceleration factors; for an axial acquisition such as in the case of this study the phase direction can be run right-left (with the arms up if outer-volume suppression techniques are not implemented) to allow better utilization of the coil array geometry and acceleration factors of at least 3 if not higher.

As a limitation of this study, the authors need to state that they did not utilize variable refocussing flip angles with SSFSE (available with some vendor implementations such as GE), and that they’re faster acquisition time was largely due to SAR constraints resulting from the constant refocusing flip angles.

Answer) Thank you for your comment. We’ve removed the inappropriate sentence and added another limitation in the manuscript.

2-18.  A new question, was the same interleaving pattern used for SSFSEconv and SSFSEcs? From the text it is unclear if the slice increment of 4 was used for just one, or both of the SSFSE sequences. This needs to be better described in the manuscript.

Answer) Slice increment of two was used for conventional SSFSE. We’ve added this information in the manuscript.

Reviewer 3 Report

Thanks for the authors' response and they resolved my concerns.

Author Response

Thank you very much.